# Machine Learning Models to Predict the Risk of Rapidly Progressive Kidney Disease and the Need for Nephrology Referral in Adult Patients with Type 2 Diabetes

**DOI:** 10.3390/ijerph20043396

**Published:** 2023-02-15

**Authors:** Chia-Tien Hsu, Kai-Chih Pai, Lun-Chi Chen, Shau-Hung Lin, Ming-Ju Wu

**Affiliations:** 1Division of Nephrology, Department of Internal Medicine, Taichung Veterans General Hospital, Taichung 40705, Taiwan; 2School of Medicine, National Yang Ming Chiao Tung University, Taipei 112304, Taiwan; 3College of Engineering, Tunghai University, Taichung 407224, Taiwan; 4DDS-THU AI Center, Tunghai University, Taichung 407224, Taiwan; 5Department of Post-Baccalaureate Medicine, College of Medicine, National Chung Hsing University, Taichung 40227, Taiwan; 6RongHsing Research Center for Translational Medicine, College of Life Sciences, National Chung Hsing University, Taichung 40227, Taiwan; 7Ph.D. Program in Translational Medicine, National Chung Hsing University, Taichung 40227, Taiwan; 8School of Medicine, Chung Shan Medical University, Taichung 40201, Taiwan; 9Graduate Institute of Biomedical Sciences, College of Medicine, China Medical University, Taichung 404333, Taiwan

**Keywords:** machine learning, type 2 diabetes, diabetic kidney disease, nephrology referral

## Abstract

Early detection of rapidly progressive kidney disease is key to improving the renal outcome and reducing complications in adult patients with type 2 diabetes mellitus (T2DM). We aimed to construct a 6-month machine learning (ML) predictive model for the risk of rapidly progressive kidney disease and the need for nephrology referral in adult patients with T2DM and an initial estimated glomerular filtration rate (eGFR) ≥ 60 mL/min/1.73 m^2^. We extracted patients and medical features from the electronic medical records (EMR), and the cohort was divided into a training/validation and testing data set to develop and validate the models on the basis of three algorithms: logistic regression (LR), random forest (RF), and extreme gradient boosting (XGBoost). We also applied an ensemble approach using soft voting classifier to classify the referral group. We used the area under the receiver operating characteristic curve (AUROC), precision, recall, and accuracy as the metrics to evaluate the performance. Shapley additive explanations (SHAP) values were used to evaluate the feature importance. The XGB model had higher accuracy and relatively higher precision in the referral group as compared with the LR and RF models, but LR and RF models had higher recall in the referral group. In general, the ensemble voting classifier had relatively higher accuracy, higher AUROC, and higher recall in the referral group as compared with the other three models. In addition, we found a more specific definition of the target improved the model performance in our study. In conclusion, we built a 6-month ML predictive model for the risk of rapidly progressive kidney disease. Early detection and then nephrology referral may facilitate appropriate management.

## 1. Introduction

Diabetes mellitus (DM) is a major cause of life expectancy reduction and premature death [1,2,3]. The mortality in diabetic patients is significantly increased when the renal function is impaired [4]. With the improvement in treatment, trends in the rates of some diabetic complications have decreased, such as stroke or acute myocardial infarction, although the burden of diabetes is continuously increasing. However, diabetic kidney disease is still the leading cause of end-stage kidney disease (ESKD) [5,6]. According to the report by the United States Renal Data System (USRDS) in 2020 [7], Taiwan has persistently reported the highest incidence and prevalence of end-stage kidney disease worldwide. In the 2020 annual report on kidney disease in Taiwan, Lai et al. [8] reported the percentage of diabetes among incident dialysis patients increased from 45.3% in 2010 to 46.2% in 2018, and the percentage of diabetes among prevalent dialysis patients increased from 39.7% in 2010 to 47.8% in 2018. Early detection of rapidly progressive kidney disease and nephrology referral is an important point to decrease complications and mortality [9]. Albuminuria is an important marker of diabetic kidney disease (DKD) and is associated with a poor outcome, but some type 2 diabetes mellitus (T2DM) patients have a GFR decline before the onset of albuminuria [10]. In addition, nondiabetic kidney diseases are also the possible cause of rapidly progressive kidney disease [11], and these patients should be promptly referred to an experienced nephrologist for further surveying and management. With the heterogeneous phenotype of type 2 diabetic renal disease, novel tools are required for the early detection of rapidly progressive kidney disease and the need for nephrology referral in T2DM patients.

Artificial intelligence (AI) has been widely applied in medical fields for diagnostic assistance, outcome prediction, and guiding treatment. Machine learning (ML) is a subset of AI. ML models are algorithms that teach a computer to learn from data [12,13]. There have been some studies of AI applications in DKD [14,15,16,17,18], but only a few studies have focused on the prediction of diabetic nephropathy and renal function decline [15,17]. The aim of our study was to construct a 6-month ML predictive model for the risk of rapidly progressive kidney disease and the need for nephrology referral in adult patients with T2DM.

## 2. Materials and Methods

### 2.1. Study Subjects

We retrospectively extracted the electronic medical records (EMR) in our hospital from January 2008 to June 2021. Among them, we found 62,360 patients with a diagnosis of type 2 diabetes mellitus (T2DM) according to the International Classification of Diseases codes, and the inclusion criteria of our study were as follows: (1) hospitalized at least once with ICD-9 or ICD10 coding for T2DM, (2) at least two outpatient ICD-9 or ICD10 codings for T2DM, and (3) age at diagnosis of T2DM ≥ 20 years. The exclusion criteria were (1) patients who underwent dialysis before the diagnosis of T2DM and (2) renal transplant patients. Our study was approved by the institutional review board of Taichung Veterans General Hospital (IRB TCVGH No: SE22064A). Patient informed consent was waived because all protected health information was deidentified and the retrospective data analysis nature of this study. This research was funded by grants from the Ministry of Science and Technology of Taiwan (MOST 108-2314-B-005-005-MY3).

### 2.2. Data Extraction

All the extracted personal information of the patients was deidentified. The demographic features used for the machine learning models included age, sex, height, weight, and body mass index (BMI). The laboratory features include serum creatinine (Cr), blood urea nitrogen (BUN), fasting glucose, random glucose, glycated hemoglobin (HbA1c), spot urine protein to creatinine ratio (UPCR), spot urine albumin to creatinine ratio (UACR), hemoglobin (HGB), hematocrit (HCT), albumin, total protein, aspartate aminotransferase (AST), alanine transaminase (ALT), creatine phosphokinase (CPK), high-sensitivity C-reactive protein (hsCRP), serum sodium (Na), serum potassium (K), red blood count (RBC), white blood count (WBC), platelet, total bilirubin (Bil-T), uric acid (UA), total cholesterol (CHO), low-density lipoprotein (LDL), and triglyceride (TG). The comorbidities were extracted according to the ICD-9 or ICD-10 codes and included diabetic retinopathy, hypertension, coronary arterial disease (CAD), stroke, peripheral arterial disease (PAD), congestive heart failure (CHF), acute kidney injury (AKI), liver cirrhosis, cancer, bacteremia, sepsis, shock, peritonitis, ascites, and bleeding esophageal varices.

### 2.3. Study Design and Label Definition

In this study, we aimed to construct multiple machine learning models to predict the risk of rapidly progressive kidney disease and the need for nephrology referral in diabetes patients. We compared two different prediction outcomes of renal function deterioration and the need for nephrology referral in diabetes patients (Figure 1): (1) the estimated glomerular filtration rate (eGFR) falling below 30 mL/min/1.73 m^2^ and (2) the eGFR falling below 45 mL/min/1.73 m^2^. Clinical guidelines [19,20] recommend the referral of DM patients to nephrology when the eGFR falls below 30 mL/min/1.73 m^2^. However, a previous study showed a GFR < 45 mL/min/1.73 m^2^ at the time of referral is also a significant risk factor for mortality [21]. Hence, the outcomes of our predictive models were aggravated renal function from eGFR ≥ 60 mL/min/1.73 m^2^ to (1) GFR < 30 mL/min/1.73 m^2^ and (2) to <45 mL/min/1.73 m^2^.

We selected adult T2DM patients with pair eGFR records of a 180-day period between the reference point and prediction target point. We first determined the target point for each individual patient and then went back to determine the reference point to select patients who fitted the criteria for the reference point. We labeled patients as being in the “referral” group if the eGFR was persistently lower than our outcomes (eGFR < 45 or <30 mL/min/1.73 m^2^) at the target point and 90 days after the target point. We confirmed chronic kidney disease if the eGFR did not recover 90 days after the target point in the “referral” group. On the other hand, we labeled patients as being in the “non-referral” group if (1) the eGFR was persistently ≥ 30 mL/min/1.73 m^2^ at the target point and 90 days after the target point or (2) the eGFR was persistently ≥ 45 mL/min/1.73 m^2^ at the target point and 90 days after the target point. We further enrolled patients according to the criteria for the reference point as follows: (1) eGFR ≥ 60 mL/min/1.73 m^2^ at the reference point, (2) 180-day average eGFR ≥ 60 mL/min/1.73 m^2^ prior to the reference point, and (3) T2DM diagnosis before the reference point.

### 2.4. Data Preprocessing and Machine Learning Models

We discussed with the domain experts for outliers of laboratory features. We excluded outliers of laboratory features on the basis of medical knowledge, wherein the error values were obviously inconsistent with the actual situation. Patients in the non-referral group had a more stable condition than patients in the referral group, which resulted in less laboratory examinations among patients in the non-referral group. There were a few patients with more than 12 missing features in the referral group. We excluded patients with more than 12 missing features to deal with the missing data in the non-referral group and the imbalance of the data set. After that, features with more than 40% missing values were excluded, and the mean of this feature was used to interpolate the remaining missing data [22,23]. We chose the “last” and “average” values of each feature in the 180-day period before the reference point as input data (Figure 1). We treated our prediction of referral need as a binary classification problem.

The architecture of our prediction models is shown in Figure 2. The study cohort was divided into the following two parts: (1) the data from January 2008 to December 2019 as the training/validation data set, and (2) the data from January 2020 to June 2021 as the testing data set. Then, the training/validation data set was randomly divided, with 80% used for training and 20% for validation. We performed fivefold cross-validation within the training/validation data set to identify the optimal classifier [24,25,26,27,28]. The optimal classifier was then used to predict our outcome for each patient in the testing data set. The testing data set was independent of the training/validation data set. It provided an unbiased final model performance metric.

We compared the performance of three classical machine learning algorithms: logistic regression (LR), random forest (RF), and extreme gradient boosting (XGBoost) to develop the predictive models. We further applied an ensemble approach using soft voting classifier to classify the referral group [29,30]. In the ensemble model, LR, RF, and XGBoost classifier were ensembled. We used the soft voting calculated on the predicted probability of the output class. All analyses were performed using Python (version 3.8) [31]. We used the area under the receiver operating characteristic curve (AUROC), precision, recall, and accuracy as the metrics to evaluate the performance between different models. We also calculated the Shapley additive explanations (SHAP) values to evaluate the feature importance that explored the relationship between the outcome and the feature [32,33].

The assessment of normality was conducted using the Kolmogorov–Smirnov test. The continuous variables with normal distribution are shown as mean ± standard deviation, whereas the continuous variables with non-normal distribution are presented as the median (first quartile, third quartile). The categorical variables are reported as numbers (percentage). Tests for the statistical significance were conducted using the chi-squared test for categorical variables and the Mann–Whitney test for non-parametric continuous variables. The level of significance was set at *p* < 0.05. Statistical analyses were performed using MedCalc for Windows, version 20.210 (MedCalc Software, Ostend, Belgium). 

Medical data are usually unbalanced. Because the imbalance of data were found, we performed a pilot experiment with a new target of outcome (persistent eGFR ≥ 60 mL/min/1.73 m^2^) to find the optimal method for this problem (see Appendix A). We applied Downsample, the Synthetic Minority Oversampling Technique (SMOTE) algorithm, and Tomek Link [34] to cope with the imbalance of data [35,36]. However, the result (see Table A1 in Appendix A) showed no obvious improvement of performance. Finally, we input the original data set into our machine learning models without any of the abovementioned methods.

## 3. Results

A total of 19,892 adult T2DM patients were enrolled in “experiment 1” to predict the rapid renal function decline and nephrology referral when the eGFR was persistently lower than 30 mL/min/1.73 m^2^. Among these, there were 19,244 adult T2DM patients in the “non-referral” group and 648 adult T2DM patients in the “referral” group.

In addition, a total of 16,145 adult T2DM patients were enrolled in “experiment 2” to predict the rapid renal function decline and nephrology referral when the eGFR was persistently lower than 45 mL/min/1.73 m^2^. Among these, there were 15,159 adult T2DM patients in the “non-referral” group and 986 adult T2DM patients in the “referral” group.

### 3.1. Experiment 1: Predict Rapidly Progressive Kidney Disease and Nephrology Referral When the eGFR Was Persistently Lower than 30 mL/min/1.73 m^2^

Table 1 reveals the baseline demographic and clinical characteristics of the included patients in experiment 1. The age of the patients was significantly older in the referral group. Patients in the referral group had significantly more comorbidities, higher creatinine, higher BUN, higher HbA1c, lower HGB, lower albumin, higher hsCRP, higher uric acid, higher TG, higher UPCR, and higher UACR. The missing data for each variable in the experiment 1 are shown in Appendix B Table A2.

Table 2 demonstrates the three models to predict rapidly progressive kidney disease and nephrology referral when the eGFR was persistently < 30 mL/min/1.73 m^2^. All three models achieved an accuracy of more than 0.91 and an AUROC of more than 0.96. The XGB model had higher accuracy and relatively higher precision in the referral group as compared with the LR and RF models. However, LR and RF models had higher recall in the referral group. In general, the ensemble voting classifier had relatively higher accuracy, higher AUROC, and higher recall in the referral group as compared with the other three models.

Figure 3 shows the confusion matrix and predictive probabilities of the XGBoost model in experiment 1. The plot of the predictive probabilities in Figure 3 revealed this model could distinguish the “referral” from the “non-referral” group in both the training/validation data set (Figure 3A) and testing data set (Figure 3B).

Figure 4 demonstrates the SHAP summary plot of the top 15 features for the XGBoost model in experiment 1. The higher the SHAP value of a feature, the higher the probability of rapidly progressive kidney disease. A dot denotes each feature value for the model of each patient. The dots are colored according to the values of the features for the respective patient and accumulate to describe the density. Blue represents the lower feature values, and red represents the higher feature values.

### 3.2. Experiment 2: Predict Rapidly Progressive Kidney Disease and Nephrology Referral When the eGFR Was Persistently Lower than 45 mL/min/1.73 m^2^

Table 3 shows the baseline demographic and clinical characteristics of the included patients in experiment 2. The age of the patients was significantly older in the referral group. Patients in the referral group had significantly more comorbidities, higher creatinine, higher BUN, lower HGB, lower albumin, higher hsCRP, higher uric acid, higher TG, higher UPCR, and higher UACR. The missing data for each variable in experiment 2 is shown in Appendix B Table A3.

Table 4 reveals the three models to predict rapidly progressive kidney disease and nephrology referral when the eGFR was persistently < 45 mL/min/1.73 m^2^. All three models achieved an accuracy of more than 0.88 and an AUROC more than 0.93. The XGB model had higher accuracy and relatively higher precision in the referral group as compared with the LR and RF models. However, LR and RF models had higher recall in the referral group. In general, the ensemble voting classifier had relatively higher accuracy, higher AUROC, and higher recall in the referral group as compared with the other models.

Figure 5 shows the confusion matrix and predictive probabilities of the XGBoost model in experiment 2. The plot of the predictive probabilities of Figure 5 revealed this model could distinguish the “referral” from the “non-referral” group in both the training/validation data set (Figure 5A) and the testing data set (Figure 5B).

Figure 6 demonstrates the SHAP summary plot of the top 15 features for the XGBoost model in experiment 2. The first three features were the same in both experiment 1 and experiment 2, and the importance of proteinuria increased in experiment 2 (eGFR was persistently < 45 mL/min/1.73 m^2^) as compared with experiment 1 (eGFR was persistently < 30 mL/min/1.73 m^2^). Proteinuria (UPCR or UACR) is also an important predictor for the risk of rapidly progressive kidney disease and the need for nephrology referral.

### 3.3. Additional Experiment with Loose Inclusion and Labeling Criteria for Both Experiments 1 and 2

We conducted an additional experiment with loose inclusion and labeling criteria for both experiments 1 and 2. In this additional experiment, we included T2DM patients with one laboratory result showing an eGFR ≥ 60 mL/min/1.73 m^2^ at the reference point and a T2DM diagnosis before the reference point. We also labeled patients with only one laboratory result, showing an eGFR < 30 mL/min/1.73 m^2^ for experiment 1 and an eGFR < 45 mL/min/1.73 m^2^ for experiment 2 in this additional experiment. We did not confirm patients with a 180-day average eGFR ≥ 60 mL/min/1.73 m^2^ prior reference point and a persistently lower eGFR 90 days after the target point in this additional experiment. Table 5 reveals that the accuracy and AUROC decreased in all of the three ML models for the additional experiment with loose inclusion and labeling criteria.

## 4. Discussion

Due to the heterogeneous phenotype of type 2 diabetic renal disease, the optimal time for the nephrology referral of T2DM patients is still challenging [6]. The American Diabetes Association (ADA) recommends that (1) diabetes patients should be referred for evaluation for RRT if they have an eGFR < 30 mL/min/1.73 m^2^, and (2) diabetes patients should be referred to a physician experienced in the care of kidney disease for uncertainty about the etiology of kidney disease, difficult management issues, and rapidly progressive kidney disease [19]. However, Pinier et al. [21] performed a retrospective survival analysis in DM patients in a 13-year period, and the study showed that both an eGFR < 30 mL/min/1.73 m^2^ and <45 mL/min/1.73 m^2^ at the time of referral were powerful risk factors for mortality. Therefore, we performed one experiment with a predictive target of an eGFR < 30 mL/min/1.73 m^2^ and another one with a predictive target of eGFR < 45 mL/min/1.73 m^2^. In addition, our study design also predicted rapidly progressive kidney disease in a 6-month period. This is an important indication for nephrology referral in T2DM as well.

Few studies have focused on the prediction of diabetic nephropathy and renal function decline [14,15,16,17,18]. Makino et al. [17] constructed a logistic regression ML learning model based on big data from the electronic medical records (EMR) of diabetes patients. Their logistic regression model had 3073 features with time series data. The accuracy of their logistic regression ML model to predict DKD aggravation was 0.71. Dong et al. [15] built up a 3-year DKD risk predictive model in patients with T2DM and normo-albuminuria, and their study showed the LightGBM model was the best model with an area under curve (AUC) of 0.815. Owing to the different study design and predictive target, all models in our study achieved an accuracy of more than 0.88 and an AUROC more than 0.93. Our study mainly focused on T2DM patients with rapidly progressive kidney disease in the 6-month period, as this condition is an important indication for nephrology referral. Early detection of this condition is a key to improving renal outcome and reducing complications. Additionally, our study design confirmed the target condition with persistent renal function impairment 90 days after the target point. The more specific and strict definition of the predictive target could improve the model performance in our study (Table 5).

Our result showed that the XGB model had higher accuracy and relatively higher precision in the referral group as compared with the LR and RF models, but LR and RF models had higher recall in the referral group. The lower precision means that the model had more false alarms, and the false alarms may increase the clinical load of the nephrologist. However, the higher recall may be more important for patient safety because it means that less patients who need nephrology referral (adult T2DM patients with rapidly progressive kidney disease) are neglected. In general, the ensemble voting classifier had relatively higher accuracy, higher AUROC, and higher recall in the referral group as compared with the other three models.

Some potential limitations of this study should be acknowledged. First, the nature of the retrospective study may cause some unrecognized confounding factors to bias the findings. Second, we did not analyze the impact of medication in our study, and some medication may be associated with rapidly progressive kidney disease. Third, our study included a small sample size and was conducted at a single hospital. The majority of the population was Taiwanese. Fourth, the data set was highly unbalanced, despite our attempts to deal with this problem. Models trained on imbalanced data may cause the accuracy paradox. Precision and recall may be better metrics in such conditions. Fifth, we excluded patients with more than 12 missing features to deal with the missing data in the non-referral group and the imbalance of dataset, which may introduce bias in analysis. Sixth, only internal validation was performed in our study; external validation using a different data set is needed. Hence, further multicenter and multinational studies are required to confirm the stability of the performance of our predictive model.

## 5. Conclusions

In conclusion, we built a 6-month machine learning predictive model for the risk of rapidly progressive kidney disease and the need for nephrology referral in adult patients with T2DM and an initial eGFR ≥ 60 mL/min/1.73 m^2^. Our result showed that the XGB model had higher accuracy and relatively higher precision in the referral group as compared with the LR and RF models, but LR and RF models had higher recall in the referral group. In general, the ensemble voting classifier had relatively higher accuracy, higher AUROC, and higher recall in the referral group as compared with the other three models. Early detection of rapidly progressive kidney disease is key to improving the renal outcome and reducing complications in adult patients with T2DM.

## Figures and Tables

**Figure 1 ijerph-20-03396-f001:**
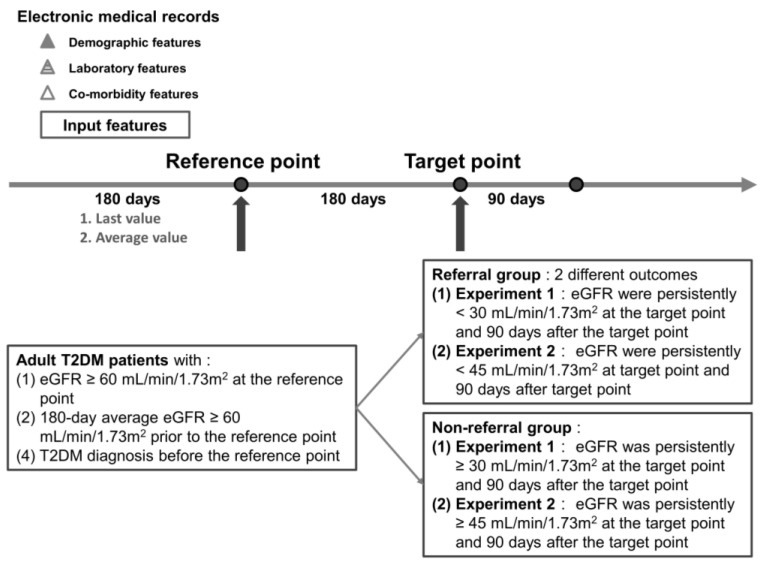
Time frame of our study design and label definition.

**Figure 2 ijerph-20-03396-f002:**
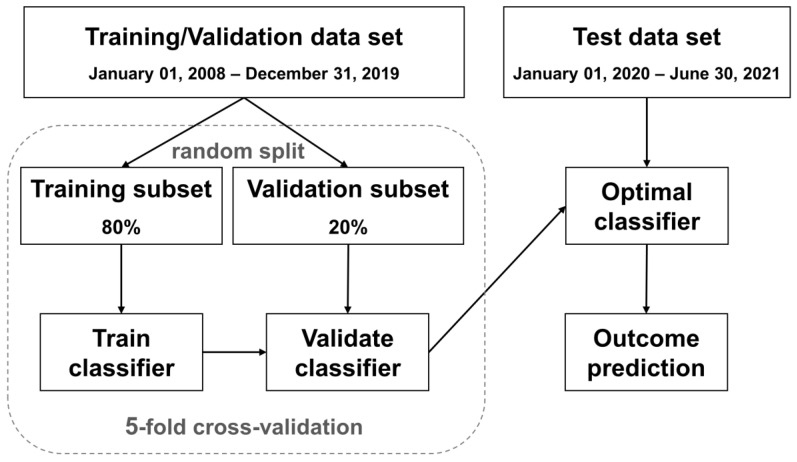
The architecture of our prediction models.

**Figure 3 ijerph-20-03396-f003:**
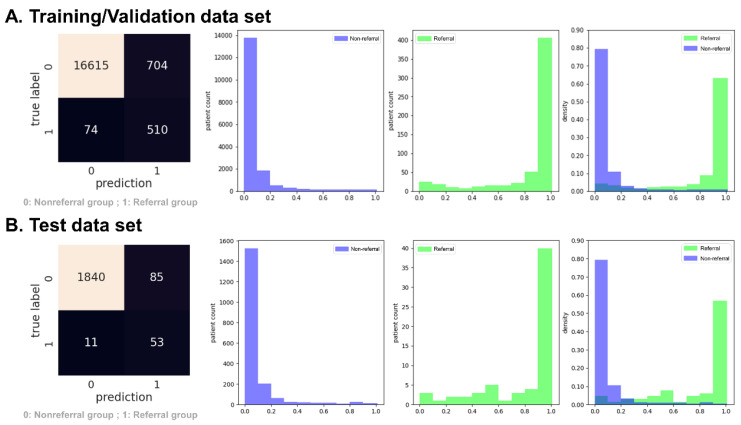
Confusion matrix and predictive probabilities histogram of the XGBoost model in experiment 1 (persistent eGFR < 30 mL/min/1.73 m^2^). The green in the histogram represents the referral group, and the medium slate blue represents the non-referral group.

**Figure 4 ijerph-20-03396-f004:**
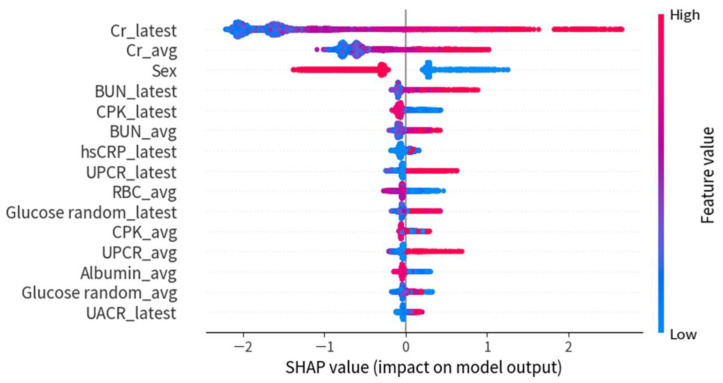
SHAP summary plot of the top 15 features for the XGBoost model in experiment 1.

**Figure 5 ijerph-20-03396-f005:**
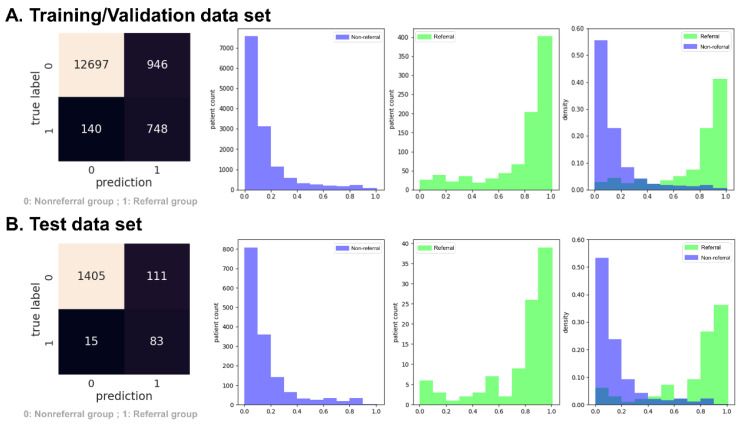
Confusion matrix and predictive probabilities histogram of the XGBoost model in experiment 2 (persistent eGFR < 45 mL/min/1.73 m^2^). The green in the histogram represents the referral group, and the medium slate blue represents the non-referral group.

**Figure 6 ijerph-20-03396-f006:**
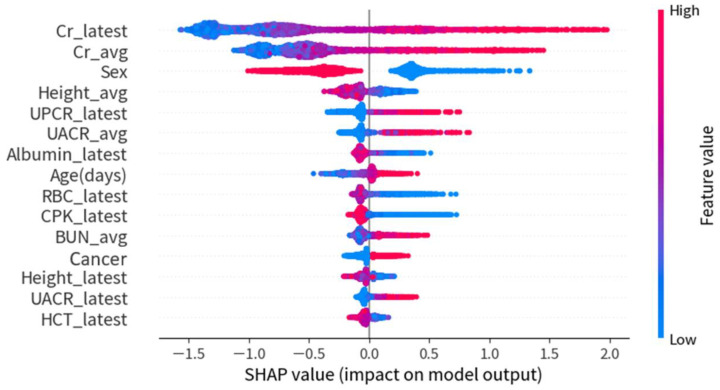
SHAP summary plot of the top 15 features of the XGBoost model in experiment 2.

**Table 1 ijerph-20-03396-t001:** Baseline demographic and clinical characteristics of the included patients in experiment 1.

	Overall (*n* = 19,892)		Referral Group (*n* = 648)		Non-Referral Group (*n* = 19,244)		*p*-Value
Age (years)	64.21	(55.85–72.55)	68.75	(59.92–76.80)	64.04	(55.70–72.30)	<0.001
Male sex	11,765	(59.14%)	391	(60.34%)	11,374	(59.10%)	0.529
Weight (kg)	66.50	(58.0–76.0)	66.0	(59.0–75.0)	66.5	(58.0–76.0)	0.904
Height (cm)	162.0	(155.5–168.0)	161.50	(155.9–168.0)	162.00	(155.5–168.0)	0.188
Hypertension	10,273	(51.64%)	464	(71.60%)	9809	(50.97%)	<0.001
CAD	3196	(16.07%)	139	(21.45%)	3057	(15.89%)	<0.001
Stroke	2962	(14.89%)	133	(20.52%)	2829	(14.70%)	<0.001
PAD	351	(1.76%)	36	(5.56%)	315	(1.64%)	<0.001
CHF	1147	(5.77%)	94	(14.51%)	1053	(5.47%)	<0.001
AKI	454	(2.28%)	103	(15.9%)	351	(1.82%)	<0.001
Liver cirrhosis	966	(4.86%)	42	(6.48%)	924	(4.80%)	0.050
Cancer	5480	(27.55%)	138	(21.30%)	5342	(27.76%)	<0.001
Bacteremia	644	(3.24%)	39	(6.02%)	605	(3.14%)	<0.001
Shock	289	(1.45%)	15	(2.31%)	274	(1.42%)	0.060
Peritonitis	282	(1.42%)	8	(1.23%)	274	(1.42%)	0.689
Ascites	167	(0.84%)	10	(1.54%)	157	(0.82%)	0.046
EV bleeding	44	(0.22%)	1	(0.15%)	43	(0.22%)	0.713
Creatinine (mg/dl)	0.89	(0.72–1.10)	1.75	(1.48–2.05)	0.88	(0.71–1.07)	<0.001
BUN (mg/dl)	16.0	(13.0–21.0)	28.0	(22.0–35.0)	16.0	(12.5–20.0)	<0.001
Fasting glucose (mg/dl)	123.0	(105.0–147.0)	123.0	(103.0–161.0)	123.0	(105.0–146.0)	<0.001
Random glucose (mg/dl)	136.0	(111.0–180.0)	143.0	(113.0–197.0)	135.5	(111.0–180.0)	0.019
HbA1c (%)	6.8	(6.2–7.7)	7.1	(6.3–8.0)	6.8	(6.2–6.7)	<0.001
HGB (g/dL)	13.3	(11.8–14.6)	11.4	(10.0–12.8)	13.4	(11.9–14.7)	<0.001
HCT (%)	39.29	(34.80–43.00)	34.50	(30.33–38.50)	39.45	(35.10–43.10)	<0.001
Albumin (g/dL)	4.10	(3.60–4.40)	3.80	(3.40–4.10)	4.10	(3.60–4.40)	<0.001
AST (U/L)	25.0	(19.0–36.0)	23.3	(17.0–33.0)	25.0	(19.0–36.0)	<0.001
ALT (U/L)	24.0	(17.0–37.0)	20.0	(13.0–28.0)	24.0	(17.0–37.0)	<0.001
CPK (U/L)	87.0	(54.0–144.0)	88.0	(54.3–165.8)	87.0	(53.6–143.0)	0.1415
hsCRP (mg/dl)	0.66	(0.15–3.80)	1.29	(0.23–4.69)	0.64	(0.15–3.77)	<0.001
K (mEq/L)	4.10	(3.80–4.40)	4.30	(3.90–4.61)	4.10	(3.80–4.40)	<0.001
RBC (×10^6^/μL)	4.32	(3.83–4.78)	3.81	(3.28–4.32)	4.33	(3.85–4.79)	<0.001
WBC (/μL)	7320	(5808–9350)	7310	(5800–9350)	7510	(6015–9376)	0.282
Bil-T (mg/dl)	0.60	(0.40–0.80)	0.50	(0.30–0.70)	0.60	(0.40–0.80)	<0.001
Uric acid (mg/dl)	5.9	(4.9–7.9)	6.8	(5.4–8.2)	5.9	(4.9–7.0)	<0.001
CHO (mg/dl)	163.0	(140.0–190.9)	160.0	(130.0–191.8)	163.0	(140.0–190.0)	0.568
LDL (mg/dl)	93.0	(74.0–117.0)	93.0	(74.0–115.0)	93.0	(74.0–117.0)	0.9311
TG (mg/dl)	122.0	(86.0–174.0)	139.0	(98.5–201.3)	121.0	(86.0–173.0)	<0.001
UPCR (mg/g)	140.0	(79.5–380.0)	1180.0	(317.5–3318.0)	120.0	(71.0–270.0)	<0.001
UACR (mg/g)	16.1	(6.9–61.8)	696.9	(118.5–2106.0)	14.9	(6.7–51.3)	<0.001

Values are expressed as median (interquartile range) or number (percentage). Non-normally distributed continuous variables were compared using the Mann–Whitney test. Categorical variables were compared using the chi-squared test. The *p*-value represents the comparison between the referral group and the non-referral group. CAD, coronary arterial disease; PAD, peripheral arterial disease; CHF, congestive heart failure; AKI, acute kidney injury; EV, esophageal varices; BUN, blood urea nitrogen; HbA1c, glycated hemoglobin; HGB, hemoglobin; HCT, hematocrit; AST, aspartate amino transferase; ALT, alanine transaminase; CPK, creatine phosphokinase; hsCRP, high-sensitivity C-reactive protein; K, serum potassium; RBC, red blood count; WBC, white blood count; Bil-T, total bilirubin; CHO, total cholesterol; LDL, low-density lipoprotein; TG, triglyceride; UPCR, spot urine protein to creatinine ratio; UACR, spot urine albumin to creatinine ratio.

**Table 2 ijerph-20-03396-t002:** Performance metrics for the three models to predict rapidly progressive kidney disease and nephrology referral when the eGFR was persistently <30 mL/min/1.73 m^2^.

Models	Data Sets	Accuracy	Referral Group (*n* = 648)		Non-Referral Group (*n* = 19,244)		AUROC
			Recall	Precision	Recall	Precision	
XGB	Validation	0.96 ± 0.01	0.87 ± 0.04	0.42 ± 0.02	0.96 ± 0.00	1.00 ± 0.00	0.97 ± 0.01
	Test	0.95 ± 0.00	0.80 ± 0.02	0.37 ± 0.01	0.95 ± 0.00	0.99 ± 0.00	0.96 ± 0.00
LR	Validation	0.94 ± 0.00	0.91 ± 0.03	0.34 ± 0.02	0.94 ± 0.01	1.00 ± 0.00	0.97 ± 0.01
	Test	0.93 ± 0.00	0.87 ± 0.01	0.30 ± 0.01	0.93 ± 0.00	1.00 ± 0.00	0.97 ± 0.00
RF	Validation	0.91 ± 0.01	0.91 ± 0.02	0.26 ± 0.02	0.91 ± 0.01	1.00 ± 0.00	0.97 ± 0.01
	Test	0.91 ± 0.00	0.91 ± 0.01	0.26 ± 0.00	0.91 ± 0.00	1.00 ± 0.00	0.96 ± 0.00
Voting	Validation	0.95 ± 0.00	0.91 ± 0.03	0.37 ± 0.02	0.95 ± 0.00	1.00 ± 0.00	0.98 ± 0.01
	Test	0.94 ± 0.00	0.86 ± 0.01	0.33 ± 0.01	0.94 ± 0.00	1.00 ± 0.00	0.97 ± 0.00

AUROC, area under the receiver operating characteristic curve; XGBoost, extreme gradient boosting; LR, logistic regression; RF, random forest; Voting, ensemble voting classifier.

**Table 3 ijerph-20-03396-t003:** Baseline demographic and clinical characteristics of the included patients in experiment 2.

	Overall (*n* = 16,145)		Referral Group (*n* = 986)		Non-Referral Group (*n* = 15,159)		*p*-Value
Age (years)	63.88	(55.62–72.05)	69.84	(61.79–78.05)	62.46	(54.37–70.13)	<0.001
Male sex	9425	(58.38%)	563	(57.10%)	8862	(58.46%)	0.401
Weight (kg)	66.3	(58.0–75.9)	64.5	(57.0–73.1)	67.0	(58.3–76.5)	<0.001
Height (cm)	162.0	(155.5–168.0)	161.0	(154.0–166.7)	162.1	(156.0–168.5)	<0.001
Hypertension	8066	(49.96%)	662	(67.14%)	7404	(48.84%)	<0.001
CAD	2565	(15.89%)	207	(20.99%)	2358	(15.56%)	<0.001
Stroke	2256	(13.97%)	202	(20.49%)	2054	(13.55%)	<0.001
PAD	248	(1.54%)	32	(3.25%)	216	(1.42%)	<0.001
CHF	786	(4.87%)	132	(13.39%)	654	(4.31%)	<0.001
AKI	165	(1.02%)	79	(8.01%)	86	(0.57%)	<0.001
Liver cirrhosis	748	(4.63%)	62	(6.29%)	686	(4.53%)	0.011
Cancer	4345	(26.91%)	279	(28.30%)	4066	(26.82%)	0.312
Bacteremia	426	(2.64%)	52	(5.27%)	374	(2.47%)	<0.001
Shock	179	(1.11%)	23	(2.33%)	156	(1.03%)	<0.001
Peritonitis	211	(1.31%)	22	(2.23%)	189	(1.25%)	0.008
Ascites	114	(0.71%)	17	(1.72%)	97	(0.64%)	<0.001
EV bleeding	37	(0.23%)	4	(0.41%)	33	(0.22%)	0.232
Creatinine (mg/dl)	0.88	(0.71–1.07)	1.14	(0.94–1.38)	0.83	(0.70–0.99)	<0.001
BUN (mg/dl)	16.0	(12.3–20.0)	19.5	(15.0–25.0)	15.0	(12.0–18.5)	<0.001
Fasting glucose (mg/dl)	123.0	(106.0–147.0)	124.0	(104.6–153.0)	123.0	(106.0–146.0)	0.277
Random glucose (mg/dl)	136.0	(111.0–180.6)	142.0	(112.0–192.0)	134.0	(111.0–177.0)	<0.001
HbA1c (%)	6.8	(6.2–7.7)	6.9	(6.2–7.8)	6.8	(6.3–7.7)	0.632
HGB (g/dL)	13.4	(11.9–14.6)	12.1	(10.7–13.5)	13.6	(12.2–14.8)	<0.001
HCT (%)	39.20	(34.70–42.90)	35.90	(31.60–40.00)	40.00	(35.90–43.45)	<0.001
Albumin (g/dL)	4.10	(3.55–4.40)	3.90	(3.37–4.20)	4.10	(3.65–4.40)	<0.001
AST (U/L)	25.0	(19.0–37.0)	25.0	(19.0–39.0)	25.0	(19.0–36.0)	0.017
ALT (U/L)	24.0	(17.0–37.0)	21.0	(15.0–34.0)	25.0	(17.0–38.0)	<0.001
CPK (U/L)	86.0	(52.0–141.0)	79.7	(47.0–138.0)	89.0	(55.0–142.0)	<0.001
hsCRP (mg/dl)	0.65	(0.15–3.74)	1.28	(0.26–5.46)	0.51	(0.12–3.16)	<0.001
K (mEq/L)	4.10	(3.80–4.40)	4.20	(3.81–4.50)	4.10	(3.80–4.40)	<0.001
RBC (×10^6^/μL)	4.31	(3.82–4.77)	3.94	(3.46–4.44)	4.41	(3.96–4.84)	<0.001
WBC (/μL)	7250	(5760–9260)	7300	(5700–9307)	7240	(5780–9250)	0.797
Bil-T (mg/dl)	0.60	(0.40–0.80)	0.54	(0.40–0.80)	0.60	(0.40–0.80)	<0.001
Uric acid (mg/dl)	5.9	(4.9–7.1)	6.4	(5.2–7.6)	5.8	(4.8–6.9)	<0.001
CHO (mg/dl)	163.0	(140.0–190.0)	157.0	(133.0–185.0)	165.0	(142.0–192.0)	<0.001
LDL (mg/dl)	93.0	(74.0–116.0)	90.0	(70.0–113.0)	94.0	(75.0–117.0)	<0.001
TG (mg/dl)	121.0	(86.0–174.0)	124.0	(88.0–179.0)	120.0	(85.0–172.0)	0.003
UPCR (mg/g)	141.0	(80.0–370.0)	200.0	(98.8–572.3)	115.0	(70.0–240.0)	<0.001
UACR (mg/g)	15.9	(6.8–60.5)	57.7	(14.7–334.5)	12.6	(6.2–38.1)	<0.001

Values are expressed as median (interquartile range) or number (percentage). Non-normally distributed continuous variables were compared using the Mann–Whitney test. Categorical variables were compared using the chi-squared test. The *p*-value represents the comparison between the referral group and the non-referral group. CAD, coronary arterial disease; PAD, peripheral arterial disease; CHF, congestive heart failure; AKI, acute kidney injury; EV, esophageal varices; BUN, blood urea nitrogen; HbA1c, glycated hemoglobin; HGB, hemoglobin; HCT, hematocrit; AST, aspartate amino transferase; ALT, alanine transaminase; CPK, creatine phosphokinase; hsCRP, high-sensitivity C-reactive protein; K, serum potassium; RBC, red blood count; WBC, white blood count; Bil-T, total bilirubin; CHO, total cholesterol; LDL, low-density lipoprotein; TG, triglyceride; UPCR, spot urine protein to creatinine ratio; UACR, spot urine albumin to creatinine ratio.

**Table 4 ijerph-20-03396-t004:** Performance metrics for the three models to predict rapidly progressive kidney disease and nephrology referral when the eGFR was persistently < 45 mL/min/1.73 m^2^.

Models	Data Sets	Accuracy	Referral Group (*n* = 986)		Non-Referral Group (*n* = 15,159)		AUROC
			Recall	Precision	Recall	Precision	
XGB	Validation	0.93 ± 0.01	0.84 ± 0.03	0.44 ± 0.02	0.93 ± 0.01	0.99 ± 0.00	0.95 ± 0.01
	Test	0.92 ± 0.00	0.84 ± 0.01	0.42 ± 0.01	0.93 ± 0.00	0.99 ± 0.00	0.93 ± 0.01
LR	Validation	0.90 ± 0.01	0.88 ± 0.04	0.37 ± 0.02	0.90 ± 0.01	0.99 ± 0.00	0.95 ± 0.01
	Test	0.90 ± 0.00	0.87 ± 0.03	0.35 ± 0.01	0.90 ± 0.00	0.99 ± 0.00	0.94 ± 0.01
RF	Validation	0.89 ± 0.01	0.89 ± 0.03	0.33 ± 0.01	0.89 ± 0.00	0.99 ± 0.00	0.95 ± 0.01
	Test	0.88 ± 0.00	0.86 ± 0.03	0.32 ± 0.01	0.88 ± 0.00	0.99 ± 0.00	0.93 ± 0.00
Voting	Validation	0.91 ± 0.00	0.88 ± 0.02	0.40 ± 0.01	0.91 ± 0.01	0.99 ± 0.00	0.95 ± 0.01
	Test	0.91 ± 0.00	0.87 ± 0.02	0.39 ± 0.01	0.91 ± 0.00	0.99 ± 0.00	0.94 ± 0.00

AUROC, area under the receiver operating characteristic curve; XGBoost, extreme gradient boosting; LR, logistic regression; RF, random forest; Voting, ensemble voting classifier.

**Table 5 ijerph-20-03396-t005:** Performance metrics for the three models with loose inclusion and labeling criteria compared with experiment 1 and experiment 2.

Loose Inclusion and Labeling Criteria Compared with Experiment 1							
Models	Data Sets	Accuracy	Referral Group (*n* = 2541)		Non-Referral Group (*n* = 19,244)		AUROC
			Recall	Precision	Recall	Precision	
XGB	Validation	0.87 ± 0.00	0.78 ± 0.02	0.46 ± 0.01	0.88 ± 0.01	0.97 ± 0.00	0.92 ± 0.00
	Test	0.87 ± 0.00	0.80 ± 0.01	0.47 ± 0.00	0.88 ± 0.00	0.97 ± 0.00	0.92 ± 0.00
LR	Validation	0.85 ± 0.01	0.81 ± 0.01	0.42 ± 0.01	0.85 ± 0.01	0.97 ± 0.00	0.91 ± 0.01
	Test	0.84 ± 0.00	0.80 ± 0.01	0.41 ± 0.00	0.85 ± 0.00	0.97 ± 0.00	0.90 ± 0.00
RF	Validation	0.83 ± 0.00	0.83 ± 0.03	0.40 ± 0.01	0.83 ± 0.01	0.97 ± 0.00	0.92 ± 0.01
	Test	0.84 ± 0.00	0.84 ± 0.01	0.41 ± 0.01	0.84 ± 0.00	0.98 ± 0.00	0.92 ± 0.00
**Loose Inclusion and Labeling Criteria Compared with Experiment 2**							
**Models**	**Data Sets**	**Accuracy**	**Referral Group** **(*n* = 3836)**		**Non-Referral Group** **(*n* = 15,159)**		**AUROC**
			**Recall**	**Precision**	**Recall**	**Precision**	
XGB	Validation	0.84 ± 0.01	0.81 ± 0.01	0.58 ± 0.01	0.85 ± 0.00	0.95 ± 0.00	0.91 ± 0.00
	Test	0.84 ± 0.00	0.79 ± 0.00	0.57 ± 0.01	0.85 ± 0.00	0.94 ± 0.00	0.90 ± 0.00
LR	Validation	0.83 ± 0.00	0.81 ± 0.01	0.55 ± 0.01	0.83 ± 0.00	0.95 ± 0.00	0.90 ± 0.01
	Test	0.83 ± 0.00	0.82 ± 0.00	0.55 ± 0.00	0.83 ± 0.00	0.95 ± 0.00	0.89 ± 0.00
RF	Validation	0.82 ± 0.01	0.83 ± 0.01	0.53 ± 0.01	0.81 ± 0.01	0.95 ± 0.00	0.90 ± 0.01
	Test	0.82 ± 0.00	0.81 ± 0.01	0.54 ± 0.01	0.83 ± 0.00	0.95 ± 0.00	0.89 ± 0.00

AUROC, area under the receiver operating characteristic curve; XGBoost, extreme gradient boosting; LR, logistic regression; RF, random forest.

## Data Availability

Not applicable.

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
