# Peer review of "Machine Learning Models to Predict the Risk of Rapidly Progressive Kidney Disease and the Need for Nephrology Referral in Adult Patients with Type 2 Diabetes"

_ijerph, 2023, doi:10.3390/ijerph20043396_

Round 1
Reviewer 1 Report
Dr. Chia-Tien et al. developed a 6-month machine learning (ML) predictive model for the risk of rapidly progressive kidney disease and the need for nephrology referral in adult patients with T2DM and an initial estimated glomerular filtration rate (eGFR) ≥ 60 mL/min/1.73m2. They have extracted patients and medical features from the electronic medical records (EMR), dividing the cohort into a training/validation and testing data set to develop and validate the models based on three algorithms: logistic regression (LR), random forest (RF), and extreme Gradient Boosting (XGBoost) with the latter achieving better predictive effects with an accuracy above 0.92 and an AUROC above 0.93.
The study is interesting however, there are significant issues that hinder the manuscript publication and need to be addressed.
Results/Discussion
Minor – Table 4, along with the presented results revolving around it should be translated to the result section
Major
1. Patient informed consent…I understand that this was a retrospective study, and according to the current guidelines referring to Informed Consent, the IRB may waive informed consent, however, is there any form of approval from the patients included in this study stating that their medical data can be further used in research?
2. Univariate analysis for patient predictors(nominal, ordinal, continuous) should be presented in a table along with the number of total cases. The missing data for each variable should also be presented likewise.
3. You should refer to age, sex, height, weight, and body mass index (BMI). The laboratory features include serum creatinine (Cr), blood urea nitrogen (BUN), fasting glucose, random glucose, glycated hemoglobin (HbA1c), spot urine protein to creatinine ratio (UPCR), spot urine albumin to creatinine ratio (UACR), hemoglobin (HGB), hematocrit (HCT), albumin, total protein, aspartate amino transferase (AST), alanine transaminase (ALT), creatine phosphokinase (CPK), high sensitivity C-reactive protein (hsCRP), serum sodium (Na), serum potassium (K), red blood count (RBC), white blood count (WBC), platelet, total bilirubin (Bil-T), uric acid (UA), total cholesterol (CHO), low-density lipoprotein (LDL), and triglyceride (TG), comorbidities like diabetic retinopathy, hypertension, coronary arterial disease (CAD), stroke, peripheral arterial disease (PAD), congestive heart failure (CHF), acute kidney injury (AKI), liver cirrhosis, cancer, bacteremia, sepsis, shock, peritonitis, ascites, and bleeding esophageal varices.
4. What percentage of missingness and what type of missingness had the dataset after all based on features and missing values (lines 120-126). For example, missing completely at random(MCAR), missing at random (MAR), and so forth. You should provide references from the literature to support your approach in this respect. Formulations such as discussion with domain experts (line 120) is not appropriate to support your research design to be published.
5. Why did you choose to perform fivefold cross-validation? Why not tenfold cross-validation?
Author Response
Response to the comments
Dear reviewer 1: 
Thank you for your detailed review. We feel that your insightful suggestions helped us in improving the manuscript. We have provided a point-by-point response to all your comments below. We revise our manuscript using a word processing program. In the revised manuscript, all the changes are highlighted using track changes to make them more visible. Your original comments are listed below followed by our response to each comment.
Reviewer 1
Dr. Chia-Tien et al. developed a 6-month machine learning (ML) predictive model for the risk of rapidly progressive kidney disease and the need for nephrology referral in adult patients with T2DM and an initial estimated glomerular filtration rate (eGFR) ≥ 60 mL/min/1.73m2. They have extracted patients and medical features from the electronic medical records (EMR), dividing the cohort into a training/validation and testing data set to develop and validate the models based on three algorithms: logistic regression (LR), random forest (RF), and extreme Gradient Boosting (XGBoost) with the latter achieving better predictive effects with an accuracy above 0.92 and an AUROC above 0.93.
The study is interesting however, there are significant issues that hinder the manuscript publication and need to be addressed.
Comment:
Results/Discussion
Minor – Table 4, along with the presented results revolving around it should be translated to the result section
Reply:
Yes. Thank you for your advice. As suggested, the Table 4 (Table 7 in our revised manuscript) and related paragraph were moved to the Results section.
(Please see Table 7 and the Result Section in our revised manuscript)
Comment:
Major
- Patient informed consent…I understand that this was a retrospective study, and according to the current guidelines referring to Informed Consent, the IRB may waive informed consent, however, is there any form of approval from the patients included in this study stating that their medical data can be further used in research?
Reply:
Yes. Thank you for your comment. We deeply appreciate your valuable opinion. The IRB approved our request to waive the documentation of informed consent according to the U.S Department of Health & Human Services (HHS) regulations. Federal regulation and FDA Guidance establishes five criteria for waiving consent or altering the elements of consent in minimal risk studies according to 45 CFR 46.116(f) (Common Rule). One of the five criteria is "The research involves no more than minimal risk". Minimal risk means that the probability and magnitude of harm or discomfort anticipated in the research are not greater than those ordinarily encountered in daily life or during the performance of routine physical or psychological examinations or tests (45.CFR.46.102(j)) (Common Rule). For instance, MIMIC-III (‘Medical Information Mart for Intensive Care’) is a famous database for research. Requirement for individual patient consent was also waived in the MIMIC-III project. (Reference: Johnson, A., Pollard, T., Shen, L. et al. MIMIC-III, a freely accessible critical care database. Sci Data 3, 160035 (2016).)
Our research did not influence the original patient care due to the retrospective data analysis nature of this study, and our research involves no more than minimal risk.
We revised the related sentence to “Patient informed consent was waived because all protected health information was deidentified and the retrospective data analysis nature of this study.”
(Please see the 2.1. Study subjects of Materials and Methods Section in our revised manuscript)
Comment:
- Univariate analysis for patient predictors (nominal, ordinal, continuous) should be presented in a table along with the number of total cases. The missing data for each variable should also be presented likewise.
Reply:
Yes. Thank you for your advice. We deeply appreciate your valuable opinion. As suggested, we present the baseline demographic, clinical characteristics, and the missing data in our Table 1, Table 2, Table 4, and Table 5 in our revised manuscript.
(Please see the Table 1, Table 2, Table 4, and Table 5 in our revised manuscript)
Comment:
- You should refer to age, sex, height, weight, and body mass index (BMI). The laboratory features include serum creatinine (Cr), blood urea nitrogen (BUN), fasting glucose, random glucose, glycated hemoglobin (HbA1c), spot urine protein to creatinine ratio (UPCR), spot urine albumin to creatinine ratio (UACR), hemoglobin (HGB), hematocrit (HCT), albumin, total protein, aspartate amino transferase (AST), alanine transaminase (ALT), creatine phosphokinase (CPK), high sensitivity C-reactive protein (hsCRP), serum sodium (Na), serum potassium (K), red blood count (RBC), white blood count (WBC), platelet, total bilirubin (Bil-T), uric acid (UA), total cholesterol (CHO), low-density lipoprotein (LDL), and triglyceride (TG), comorbidities like diabetic retinopathy, hypertension, coronary arterial disease (CAD), stroke, peripheral arterial disease (PAD), congestive heart failure (CHF), acute kidney injury (AKI), liver cirrhosis, cancer, bacteremia, sepsis, shock, peritonitis, ascites, and bleeding esophageal varices.
Reply:
Yes. Thank you for your advice. We deeply appreciate your valuable opinion. As suggested, we present these variables and the portion of missing data in our Table 1, Table 2, Table 4, and Table 5 in our revised manuscript.
(Please see the Table 1, Table 2, Table 4, and Table 5 in our revised manuscript)
Comment:
- What percentage of missingness and what type of missingness had the dataset after all based on features and missing values (lines 120-126). For example, missing completely at random (MCAR), missing at random (MAR), and so forth. You should provide references from the literature to support your approach in this respect. Formulations such as discussion with domain experts (line 120) is not appropriate to support your research design to be published.
Reply:
Yes. Thank you for your comment. We deeply appreciate your valuable opinion. We apologize that we made a mistake, and we will revise the related paragraph. We discussed with the domain experts for outliers of laboratory features. We excluded outliers of laboratory features based on medical knowledge, where the error values were obviously inconsistent with the actual situation.
Table 2 and Table 5 in our revised manuscript showed the non-referral group has significantly more missing data of some laboratory features. Patients in the non-referral group have more stable condition than patients in the referral group, which results in less laboratory examinations among patients in the non-referral group. The type of missingness of these laboratory variables are missing at random (MAR). Listwise deletion (or case deletion) also one of the methods for handling missing data (Please see the reference). Only few patients with more than 12 missing features were noted in the referral group, but many patients with more than 12 missing features were noted in the non-referral group. In addition, the dataset is highly unbalanced, we deleted patients with more than 12 missing features to deal with the imbalance of dataset, too. However, case deletion may result in biasness. We will mention this problem in the limitation paragraph of discussion section.
Reference:
- Soley-Bori M. Dealing with missing data: Key assumptions and methods for applied analysis. Boston Univ., 4 (2013), pp. 1-19
- Emmanuel, T., Maupong, T., Mpoeleng, D. et al. A survey on missing data in machine learning. J Big Data 8, 140 (2021). https://doi.org/10.1186/s40537-021-00516-9
As suggested, we revised the related paragraph. We discussed with the domain experts for outliers of laboratory features. We excluded outliers of laboratory features based on medical knowledge, where the error values were obviously inconsistent with the actual situation. Patients in the non-referral group have more stable condition than patients in the referral group, which results in less laboratory examinations among patients in the non-referral group. There were few patients with more than 12 missing features in the referral group. We excluded patients with more than 12 missing features to deal with the missing data in the non-referral group and the im-balance of dataset. After that, features with more than 40% missing values were excluded, and the mean of this feature was used to interpolate the remaining missing data.
(Please see 2.4. Data preprocessing and machine learning models of Methods section in our revised manuscript)
We excluded patients with more than 12 missing features to deal with the missing data in the non-referral group and the imbalance of dataset, which may introduce bias in analysis.
(Please see limitation paragraph of Discussion section in our revised manuscript)
Comment:
- Why did you choose to perform fivefold cross-validation? Why not tenfold cross-validation?
Reply:
Yes. Thank you for your comment. We deeply appreciate your valuable opinion. K fold cross validation guarantees that the score of our model does not depend on the way we picked the train and test set. Most of previous studies used values of k are five (5) or ten (10), as these two values are believed to give test error rate estimates that suffer neither from extremely high bias nor very high variance. Several previous medical studies applied fivefold cross-validation in their machine learning models. We performed fivefold cross-validation according to these references. As suggested, we added these studies in our reference list.
- A. Kumari and R. Chitra, “Classification of diabetes disease using support vector machine,” International Journal of Engineering Research in Africa, vol. 3, no. 2, pp. pp1797–1801, 2013.
- Yang T, Zhang L, Yi L, et al. Ensemble Learning Models Based on Noninvasive Features for Type 2 Diabetes Screening: Model Development and Validation. JMIR Med Inform. 2020;8(6):e15431. Published 2020 Jun 18. doi:10.2196/15431
- Prusty S, Patnaik S and Dash SK (2022) SKCV: Stratified K-fold cross-validation on ML classifiers for predicting cervical cancer. Front. Nanotechnol. 4:972421. doi: 10.3389/fnano.2022.972421
- Chauhan, N.K., Singh, K. Performance Assessment of Machine Learning Classifiers Using Selective Feature Approaches for Cervical Cancer Detection. Wireless Pers Commun 124, 2335–2366 (2022). https://doi.org/10.1007/s11277-022-09467-7
(Please see the Reference section in our revised manuscript)

Reviewer 2 Report
We congratulate the authors on designing the current study as it addresses an important public health issue at large.
Please see the critique and suggestions listed below:
· In study design, please clarify / elaborate “we further selected patients with pair events of a 180-day period … ”. It is difficult to interpret the study methodology in the description provided and this ambiguity affects accurate repeatability of the study conditions.
· As noted in the manuscript, the dataset is highly unbalanced. We note the attempts to “rebalance” the dataset, however, this was not successful. The unbalanced dataset was used in the machine learning models. This should be included in the study limitations.
· Additionally, because of the unbalanced dataset, the AUROC and accuracy alone do not accurately reflect the performance of the model. The discussion should include the relevance of low precision in the referral group and how this will impact the practical usability of this model.
· The conclusion should also include precision and recall of the referral group so that a more holistic understanding of the XGboost model is displayed.
Author Response
Dear reviewer 2: 
Thank you for your detailed review again. We feel that your insightful suggestions helped us in improving the manuscript. We have provided a point-by-point response to all your comments below. We revise our manuscript using a word processing program. In the revised manuscript, all the changes are highlighted using track changes to make them more visible. Your original comments are listed below followed by our response to each comment.
Reviewer 2
We congratulate the authors on designing the current study as it addresses an important public health issue at large.
Please see the critique and suggestions listed below:
Comment:
In study design, please clarify / elaborate “we further selected patients with pair events of a 180-day period … ”. It is difficult to interpret the study methodology in the description provided and this ambiguity affects accurate repeatability of the study conditions.
Reply:
Yes. Thank you for your comment. We deeply appreciate your valuable opinion. As suggested, we rewrite the related paragraph. We selected adult T2DM patients with pair eGFR records of a 180-day period between the reference point and prediction target point. We first determined the target point for each individual patient and then back to determine the reference point to select patients who fitted the criteria for the reference point. We labeled patients as the "referral" group if the eGFR was persistently lower than our outcomes (eGFR < 45 or < 30 mL/min/1.73m2) at the target point and 90 days after the target point. We confirmed chronic kidney disease if the eGFR did not recover 90 days after the target point in the "referral" group. On the other hand, we labeled patients as the "non-referral" group if (1) the eGFR was persistently ≥ 30 mL/min/1.73m2 at the target point and 90 days after the target point; or (2) the eGFR was persistently ≥ 45 mL/min/1.73m2 at the target point and 90 days after the target point, respectively. We further enrolled patients according to the criteria for the reference point as follows: (1) eGFR ≥ 60 mL/min/1.73m2 at the reference point, (2) 180-day average eGFR ≥ 60 mL/min/1.73m2 prior to the reference point, and (3) T2DM diagnosis before the reference point.
(Please see 2.3. Study design and label definition in our revised manuscript)
Comment:
As noted in the manuscript, the dataset is highly unbalanced. We note the attempts to “rebalance” the dataset, however, this was not successful. The unbalanced dataset was used in the machine learning models. This should be included in the study limitations.
Reply:
Yes. Thank you for your comment. We deeply appreciate your valuable opinion. As suggested, we mentioned this limitation in the discussion section. “The dataset is highly unbalanced despite our attempts to deal with this problem. Models trained on imbalanced data may cause the accuracy paradox. Precision and recall may be better metrics in such condition.”
(Please see the limitation paragraph of discussion section in our revised manuscript)
Comment:
Additionally, because of the unbalanced dataset, the AUROC and accuracy alone do not accurately reflect the performance of the model. The discussion should include the relevance of low precision in the referral group and how this will impact the practical usability of this model.
Reply:
Yes. Thank you for your advice. As suggested, we include the relevance of low precision in the referral group and how this will impact the practical usability of this model in the discussion section of our revised manuscript. Our result showed that XGB model has higher accuracy and relatively higher precision in the referral group as compared with the LR and RF models, but LR and RF models have higher recall in the referral group. The lower precision means that the model has more false alarms, and the false alarms may increase clinical load of nephrologist. However, the higher recall may be more important for patient safety because it means that less patients who need nephrology referral (adult T2DM patients with rapidly progressive kidney disease) are neglected. In general, the ensemble voting classifier has relatively higher accuracy, higher AUROC, and higher recall in the referral group as compared with the other three models.
(Please see the discussion section of our revised manuscript)
Comment:
The conclusion should also include precision and recall of the referral group so that a more holistic understanding of the XGboost model is displayed.
Reply:
Yes. Thank you for your advice. As suggested, we include the precision and recall of the referral group in the result and conclusion section. Our result showed that XGB model has higher accuracy and relatively higher precision in the referral group as compared with the LR and RF models, but LR and RF models have higher recall in the referral group. In general, the ensemble voting classifier has relatively higher accuracy, higher AUROC, and higher recall in the referral group as compared with the other three models.
(Please see conclusion section in our revised manuscript)

Reviewer 3 Report
In this manuscript, the authors tried to build a ML model to predict the risk of rapidly progressive kidney disease for patients with type 2 diabetes mellitus. It has great values for appropriate management for those patients who need nephrology referral. However, several issues still needs to be clarified.
1. The authors need to provide baseline demographic and clinical characterisitics of the included patients. This information would be very helpful when different studies are compared.
2. Although the authors claimed that XGB model has higher accuracy, LR and RF models have higher recall in the referrral group (Figure 2 and 3). It means that in LR and RF models less patients who need nephrology referral are neglected. In my opinion, this might be more important. What are the top features for LR and RF models? Can combined features by all three models have better prediction?
3. In Figure 1, for those patients who were enrolled at the reference point, there should be a possibility that eGFR were <30 ml/min/1.73m2 or < 45 ml/min/1.73m2 at the target point, but not persistent 90 days after target point. In this case, were those patients enrolled in the referral group or Non-referral group? Or, probably you first determined the target point for each individual patient and then back to determine the reference point to select patients who fitted the criteria for the reference point. Otherwise, I do not understand why in experiment 1 and 2 the patient numbers were different although the criteria for reference point is the same.
Author Response
Dear reviewer 3: 
Thank you for your detailed review again. We feel that your insightful suggestions helped us in improving the manuscript. We have provided a point-by-point response to all your comments below. We revise our manuscript using a word processing program. In the revised manuscript, all the changes are highlighted using track changes to make them more visible. Your original comments are listed below followed by our response to each comment.
Reviewer 3
In this manuscript, the authors tried to build a ML model to predict the risk of rapidly progressive kidney disease for patients with type 2 diabetes mellitus. It has great values for appropriate management for those patients who need nephrology referral. However, several issues still needs to be clarified.
Comment:
- The authors need to provide baseline demographic and clinical characteristics of the included patients. This information would be very helpful when different studies are compared.
Reply:
Yes. Thank you for your advice. We deeply appreciate your valuable opinion. As suggested, we present the baseline demographic and clinical characteristics in our Table 1 and Table 4 in our revised manuscript.
(Please see the Table 1 and Table 4 in our revised manuscript)
Comment:
- Although the authors claimed that XGB model has higher accuracy, LR and RF models have higher recall in the referral group (Figure 2 and 3). It means that in LR and RF models less patients who need nephrology referral are neglected. In my opinion, this might be more important. What are the top features for LR and RF models? Can combined features by all three models have better prediction?
Reply:
Yes. Thank you for your advice. We deeply appreciate your valuable opinion. As suggested, we include the relevance of higher recall and lower precision in the referral group and how this will impact the practical usability of this model in the discussion section of our revised manuscript. Our result showed that XGB model has higher accuracy and relatively higher precision in the referral group as compared with the LR and RF models, but LR and RF models have higher recall in the referral group. The lower precision means that the model has more false alarms, and the false alarms may increase clinical load of nephrologist. However, the higher recall may be more important for patient safety because it means that less patients who need nephrology referral (adult T2DM patients with rapidly progressive kidney disease) are neglected.
Besides, we also added the ensemble result using voting classifier which combined three models. The results were added in the Table 3 and Table 6 in our revised manuscript. The ensemble results were similar with three independent models.
A voting classifier is a machine learning estimator that trains various base models or estimators and predicts based on aggregating the findings of each base estimator. The aggregating criteria based on the decision of voting for each model output. We used the soft voting calculated on the predicted probability of the output class.
Voting classifier References
- Kumari, S., Kumar, D. & Mittal, M. An ensemble approach for classification and prediction of diabetes mellitus using soft voting classifier. International Journal of Cognitive Computing inEngineering2, 40–46 (2021).
- Oliveira, G. P. de, Fonseca, A., & Rodrigues, P. C. (2022). Diabetes diagnosis based on hard and soft voting classifiers combining statistical learning models. Brazilian Journal of Biometrics, 40(4), 415–427. https://doi.org/10.28951/bjb.v40i4.605
In general, the ensemble voting classifier has relatively higher accuracy, higher AUROC, and higher recall in the referral group as compared with the other three models.
(Please see result and discussion section in our revised manuscript)
Comment:
- In Figure 1, for those patients who were enrolled at the reference point, there should be a possibility that eGFR were <30 ml/min/1.73m2 or < 45 ml/min/1.73m2 at the target point, but not persistent 90 days after target point. In this case, were those patients enrolled in the referral group or Non-referral group? Or, probably you first determined the target point for each individual patient and then back to determine the reference point to select patients who fitted the criteria for the reference point. Otherwise, I do not understand why in experiment 1 and 2 the patient numbers were different although the criteria for reference point is the same.
Reply:
Yes. Thank you for your advice. We deeply appreciate your valuable opinion. As suggested, we rewrite the related paragraph. We selected adult T2DM patients with pair eGFR records of a 180-day period between the reference point and prediction target point. We first determined the target point for each individual patient and then back to determine the reference point to select patients who fitted the criteria for the reference point. We labeled patients as the "referral" group if the eGFR was persistently lower than our outcomes (eGFR < 45 or < 30 mL/min/1.73m2) at the target point and 90 days after the target point. We confirmed chronic kidney disease if the eGFR did not recover 90 days after the target point in the "referral" group. On the other hand, we labeled patients as the "non-referral" group if (1) the eGFR was persistently ≥ 30 mL/min/1.73m2 at the target point and 90 days after the target point; or (2) the eGFR was persistently ≥ 45 mL/min/1.73m2 at the target point and 90 days after the target point, respectively. We further enrolled patients according to the criteria for the reference point as follows: (1) eGFR ≥ 60 mL/min/1.73m2 at the reference point, (2) 180-day average eGFR ≥ 60 mL/min/1.73m2 prior to the reference point, and (3) T2DM diagnosis before the reference point.
(Please see 2.3. Study design and label definition in our revised manuscript)

Round 2
Reviewer 1 Report
The authors substantially improved the quality and soundness of the manuscript, properly addressing all my concerns, and they should be commended for their thoroughness. The manuscript is very interesting and should be published.
I would further recommend a minor spelling check, tables 2, 5 to be presented in the Appendix, and in regard to comment 5 (list of references) you might also consider this title:
NeamÈ›u, B.M.; Visa, G.; Maniu, I.; Ognean, M.L.; Pérez-Elvira, R.; Dragomir, A.; Agudo, M.; Șofariu, C.R.; Gheonea, M.; Pitic, A.; Brad, R.; Matei, C.; Teodoru, M.; Băcilă, C. A Decision-Tree Approach to Assist in Forecasting the Outcomes of the Neonatal Brain Injury. Int. J. Environ. Res. Public Health 2021, 18, 4807. https://doi.org/10.3390/ijerph18094807
Author Response
Response to the comments
Dear reviewer 1: 
Thank you for your detailed review. We feel that your insightful suggestions helped us in improving the manuscript. We have provided a point-by-point response to all your comments below. We revise our manuscript using a word processing program. In the revised manuscript, all the changes are highlighted using track changes to make them more visible. Your original comments are listed below followed by our response to each comment.
Reviewer 1
The authors substantially improved the quality and soundness of the manuscript, properly addressing all my concerns, and they should be commended for their thoroughness. The manuscript is very interesting and should be published.
Comment:
I would further recommend a minor spelling check, tables 2, 5 to be presented in the Appendix, and in regard to comment 5 (list of references) you might also consider this title:
NeamÈ›u, B.M.; Visa, G.; Maniu, I.; Ognean, M.L.; Pérez-Elvira, R.; Dragomir, A.; Agudo, M.; Șofariu, C.R.; Gheonea, M.; Pitic, A.; Brad, R.; Matei, C.; Teodoru, M.; Băcilă, C. A Decision-Tree Approach to Assist in Forecasting the Outcomes of the Neonatal Brain Injury. Int. J. Environ. Res. Public Health 2021, 18, 4807. https://doi.org/10.3390/ijerph18094807
Reply:
Yes. Thank you for your advice. We deeply appreciate your valuable opinion As suggested, we check the spelling again, and the Table 2 and Table 5 with related paragraph were moved to the Appendix B section (please see Appendix Table B1 and Appendix Table B2 in the revised manuscript). In addition, we also added the suggested paper in our list of reference (Please reference 26 in the revised manuscript).
(Please see reference section, Appendix Table B1 and Appendix Table B2 in the revised manuscript)
